# Distinct Oxidative Stress Adaptations Driven by the Overexpression of miR-526b, miR-655, and COX-2 in Breast Cancer

**DOI:** 10.3390/ijms26189103

**Published:** 2025-09-18

**Authors:** Reid M. Opperman, Sujit Maiti, Mousumi Majumder

**Affiliations:** Department of Biology, John R. Brodie Science Centre, Brandon University, 270 18th Street, Brandon, MB R7A6A9, Canada; oppermrm72@brandonu.ca (R.M.O.); maitis@brandonu.ca (S.M.)

**Keywords:** breast cancer, oxidative stress, hydrogen peroxide, microRNAs, miR-526b, miR-655, cyclooxygenase-2 (COX-2), RNA-seq, DNA damage

## Abstract

Oxidative stress has a dual role in breast cancer, promoting growth at moderate levels while causing cell death at higher levels, such as during therapeutic interventions that increase reactive oxygen species production. Oncogenic microRNAs miR-526b and miR-655 promote aggressive cancer traits—such as proliferation, migration, invasion, hypoxia response, cancer stem cell properties, and metastasis—via COX-2/EP4/PI3K pathways. These miRNAs and oxidative stress appear to engage in a self-amplifying loop, where miRNA overexpression increases ROS levels, and moderate oxidative stress, in turn, enhances miRNA expression—although the mechanisms are not yet fully understood. This study investigates how overexpressing miR-526b, miR-655, and COX-2 influences breast cancer cell responses to oxidative stress induced by H_2_O_2_. We examined cell viability, DNA damage, and transcriptomic changes in MCF7, MCF7-miR526b, MCF7-miR655, and MCF7-COX2 cell lines. Overexpression of COX-2 provided the most significant protection against oxidative stress, decreasing apoptosis and promoting cell cycle progression. Cells with miR-526b and miR-655 exhibited distinct yet overlapping stress responses, including decreased expression of DNA damage markers and alterations in p53 signaling. RNA-sequencing and network analyses identified hub genes involved in redox balance, immune, and metabolic pathways, which may have clinical significance (*OAS2*, *TNF*, *CACNA1C*, *CALML5*). Overall, these findings suggest that miR-526b, miR-655, and COX-2 play novel roles in promoting resistance to oxidative stress through transcriptional reprogramming in breast cancer; the identified markers could serve as potential biomarkers or therapeutic targets.

## 1. Introduction

Breast cancer continues to be the most frequently diagnosed cancer and a primary cause of cancer-related deaths among women globally [1]. Despite advances in treatment and detection, therapeutic resistance and treatment failure continue to pose clinical challenges. Growing evidence suggests an important involvement of oxidative stress in cancer, with consequences in carcinogenesis, tumorigenesis and treatment resistance. Oxidative stress occurs when reactive oxygen species (ROS) accumulate beyond normal levels, leading to both pro-tumorigenic and anti-tumorigenic effects [2,3]. Low-to-moderate levels of ROS can promote cell proliferation, migration, and survival, whereas excessive ROS induces cellular damage leading to senescence, apoptosis, or necrosis. ROS signaling is linked to redox-sensitive transcription factors, DNA damage response pathways, and alterations in cellular metabolism [4]. Breast cancers that display enhanced antioxidant capacities or that exhibit a dampened response to ROS-induced DNA damage, through the upregulation of pro-survival pathways or defects in apoptotic signaling, may evade ROS-induced toxicity. These forms of adaptations are clinically relevant in cases of therapeutic resistance, particularly in the context of therapies such as radiation or anthracyclines (e.g., doxorubicin), which exert part of their cytotoxic effects through the induction of oxidative stress [5,6].

MicroRNAs (miRNAs) have been shown to be key regulators of oxidative stress in cancer [7,8]. Our previous work has identified miR-526b and miR-655 as oncogenic miRNAs upregulated in response to cyclooxygenase-2 (COX-2) overexpression [9], which itself is a key source of intracellular hydrogen peroxide (H_2_O_2_) production. The overexpression of miR-526b and miR-655 in breast cancer was shown to promote the production of total intracellular ROS and superoxide (SO). Furthermore, exposure to modest levels of hydrogen peroxide for 24 h enhanced the expression of miR-526b and miR-655, indicating a positive feedback loop between both miRNAs and oxidative stress. Overexpression of miR-526b and miR-655 enhances aggressive cancer phenotypes such as proliferation, migration, invasion, cancer stem cell population in vitro, and tumorigenesis and metastasis in vivo through the COX-2/EP4/PI3K/Akt signaling pathways [10,11]. Based on these previous findings, we hypothesize that miR-526b, miR-655, and COX-2 overexpression may convey an innate ability for breast cancer cells to adapt to more cytotoxic levels of oxidative stress.

This study aims to investigate how the overexpression of miR-526b, miR-655, and COX-2 in breast cancer alters cellular responses to oxidative stress by assessing cell fate, DNA damage, and global transcriptional alteration via RNA-sequencing. Understanding how cancer cells adapt and respond to oxidative stress may have clinical relevance, helping to uncover crucial mechanisms of both chemoresistance and tumor progression.

## 2. Results

### 2.1. Cell Viability in miRNA Overexpressing Breast Cancer

The half-maximal inhibitory concentration (IC50) of cell viability for H_2_O_2_ in MCF7 breast cancer cells was determined to be 216 µM (Figure 1A). This concentration was utilized to assess the differential response of MCF7, MCF7-miR526b, MCF7-miR655, and MCF7-COX2 cell lines to H_2_O_2_ exposure. Cells were exposed to a single treatment of H_2_O_2_ for 48 h and then allowed a recovery period of three days, after which the treatment was removed, referred to as rest (H_2_O_2_ + Rest) (Figure 1B).

Following 48 h of H_2_O_2_ treatment, cell numbers in MCF7 were modestly reduced in comparison to time-zero cell counts, though this was not significant (Figure 1C). During this same period, the MCF7-miR526b and MCF7-COX2 cell lines showed a slight but non-significant increase in cell count, indicating minimal cell division. MCF7-miR655 cell numbers following treatment were unchanged.

Following the three-day recovery period, all cell lines showed a significant increase in cell number, indicating a recovery of cell proliferation following the removal of H_2_O_2_ (Figure 1C). At this time point, relative MCF7-miR526b and MCF7-miR655 cell counts were greater than those of MCF7, but not significantly, while MCF7-COX2 cell counts following recovery were significantly greater than all other cell lines. These findings suggest that the overexpression of miR-526b, miR-655, and COX-2 confer advantages to cell viability following oxidative stress, with COX-2 displaying the greatest capacity to recover.

The gene expression of cell-fate-regulatory genes *p21* (cell cycle arrest) and *PUMA* (apoptosis) were investigated in each condition (Figure 1D,E). In the control conditions, there was no significant difference in *p21* expression between cell lines, although expression was greatest in MCF7-miR655 and lowest in MCF7-COX2 (Figure 1D). After H_2_O_2_ treatment, *p21* expression was greatest in the parental MCF7 cell line; however, this was not significant. This trend was maintained following the recovery period as well, with expression in MCF7 significantly greater than that of MCF7-COX2.

Baseline *PUMA* expression was comparable across cell lines, with marginally higher levels in MCF7-miR526b and MCF7-miR655 than in MCF7 and MCF7-COX2 (Figure 1E). Following H_2_O_2_ treatment, expression was remade similar in MCF7, MCF7-miR526b, and MCF7-miR655, and it was lowest in MCF7-COX2, though not significantly. After the recovery period, levels were highest in MCF7 and MCF7-miR526b and lowest in MCF7-miR655 and MCF7-COX2, again without statistical significance.

p53 and p21 protein expression were assessed via Western blot (Figure 1F). In all cell lines, p53 protein expression was elevated after H_2_O_2_ exposure, with a significant increase in the case of MCF7 following the three-day rest period (Figure 1G). MCF7 and MCF7-miR526b had the most pronounced increase in p21 expression following H_2_O_2_ treatment, remaining elevated even after the rest period, significantly so in the case of MCF7 (Figure 1H). Though p21 expression was unchanged in MCF7-miR655 directly after the exposure period, expression did increase significantly following the rest period. In the case of MCF7-COX2, p21 expression increased significantly but rebounded following the recovery period, indicating a recovery in cell division capabilities.

AKT phosphorylation (Ser473) was analyzed following H_2_O_2_ treatment as a measure of pro-survival cell signaling (Figure 1F). AKT phosphorylation (p-AKT/total-AKT) was not significantly altered after H_2_O_2_ treatment in any cell line or condition (Figure 1I), although MCF7 and MCF7-COX2 did display a modest decrease following treatment and recovery. On the other hand, MCF7-miR526b and MCF7-miR655 showed a slight increase in AKT phosphorylation after treatment, which fell back down to baseline levels following three days of rest (Figure 1I).

Poly (ADP-ribose) polymerase (PARP) expression and cleavage were analyzed to investigate pro-apoptotic signaling after H_2_O_2_ treatment by comparing the relative abundance of full-length PARP (~116 kDa) to a cleaved fragment (~89 kDa) found during apoptosis (Figure 1F). The results show a significant increase in PARP cleavage in MCF7 cells following initial treatment, which remained elevated even after the recovery period (Figure 1J). Furthermore, PARP cleavage was significantly greater in MCF7 in comparison to both MCF7-miR526b and MCF7-COX2 after the initial 48 h treatment and remained significantly greater in comparison to MCF7-COX2 after the rest period. Conversely, only a modest increase in PARP cleavage was seen in MCF7-miR526b, MCF7-miR655, and MCF7-COX2 following H_2_O_2_ exposure. This indicates that apoptotic signals are more prominent in MCF7 cells compared to miRNA- and COX-2-overexpressing cells.

Additionally, the expression of genes involved in antioxidant defense, catalase (*CAT*), glutathione peroxidase 2 (*GPX2*), and glutathione reductase (*GSR*), were assessed, though differences were not significant and displayed extreme variability across biological replicates (Appendix A).

### 2.2. Assessing Oxidative-Stress-Induced DNA Damage and Repair

Overall, the differences in DNA damage and DNA damage response are subtle. Analysis of comet assays (Figure 2A), including tail DNA percentage (Figure 2B), tail moment (Figure 2C), and olive moment (Figure 2D), indicate that DNA damage increased in all cell lines following H_2_O_2_ treatment, although the changes were not significant when comparing conditions within the same cell line. For each comet metric, MCF7 was significantly greater than those of the other cell lines within the treatment condition and following the rest period, except for the comet tail percentage and olive moment of MCF7-miR526b after treatment and the comet tail percentage of MCF7-miR655 following the rest period. In MCF7-miR526b and MCF7-COX2 cells, all comet metrics decreased after the rest period.

γ-H2AX staining, which indicates DNA double-strand break (Figure 3A–D), increased significantly in all H_2_O_2_-exposed cells (Figure 3E). After rest, staining intensity significantly decreased in all cell lines, except for MCF7-miR655. While staining significantly reduced in MCF7, it remained significantly elevated relative to the untreated cells. In line with the comet assay, γ-H2AX staining suggests that DNA damage was greatest in the MCF7 cells after H_2_O_2_ treatment (Figure 3E), significantly greater than MCF7-miR655.

After H_2_O_2_ treatment, ATM protein expression did not appear to be altered in any of the cell lines (Figure 3F), other than MCF7-miR655 after the recovery period, although the baseline expression of ATM in these cells seemed to be the lowest. Quantification of ATM expression relative to the control condition was not significantly altered in any cell line (Figure 3G).

Similarly, total Chk2 protein expression (Figure 3F) was not significantly altered in any case, other than MCF7-COX2 following the recovery period, in which CHK2 expression was significantly reduced (Figure 3H). p-Chk2 increased after treatment for MCF7 and MCF7-COX2 and reduced slightly in MCF7-miR526b and MCF7-miR655 (Figure 3F,I), though neither case was statistically significant.

### 2.3. Identification of DEGs Associated with miRNA Overexpression via RNA-Sequencing

RNA-seq analysis was performed on MCF7, MCF7-miR526b, MCF7-miR655, and MCF7-COX2 following 48 h of H_2_O_2_ treatment. Differentially expressed genes (DEGs) were identified for each miRNA-overexpressing aggressive cell line MCF7-miR526b, MCF7-miR655, and MCF7-COX2 relative to the miRNA-low MCF7 cells (Figure 4A–C). Genes with a fold change above or below 1.5 and −1.5, as well as an FDR-adjusted *p*-value less than 0.05, were considered significant DEGs.

Comparison of these DEGs indicates that each experimental cell line differed in response to H_2_O_2_ relative to MCF7 (Figure 4D,E). Of the upregulated DEGs (Figure 4D), we identified 70 unique to MCF7-miR526b, 83 unique to MCF7-miR655, and 461 unique to MCF7-COX2. Ten upregulated DEGs were shared between MCF7-miR526b and MCF7-miR655, five were shared between MCF7-miR526b and MCF7-COX2, three were shared between MCF7-miR655 and MCF7-COX2, and only one was shared between all experimental cell lines relative to MCF7. Of the downregulated DEGs (Figure 4E), we identified 82 unique to MCF7-miR526b, 165 unique to MCF7-miR655, and 1742 unique to MCF7-COX2. There were 34 downregulated DEGs shared between MCF7-miR526b and MCF7-miR655, 97 between MCF7-miR526b and MCF7-COX2, 60 shared between MCF7-miR655 and MCF7-COX2, and 11 shared between all experimental cell lines relative to MCF7.

Principal component analysis (PCA) was performed to visualize global transcriptional differences across all samples (Figure 4F). Principal component 1 (PC1) and principal component 2 (PC2) captured 24.7% and 8.5% of the total variance, respectively. The analysis revealed a clear separation between cell lines, specifically with MCF7-COX2 relative to the other cell lines across PC1. MCF7-miR526b, MCF7-miR655, and MCF7-COX2 samples, both control and treated, clustered more tightly compared to MCF7, suggesting less variability following H_2_O_2_ treatment. This may suggest that MCF7 cells undergo a more pronounced or heterogeneous global transcriptomic response to H_2_O_2_. On the other hand, it may also suggest variability within the MCF7 cell line, which may reflect technical errors in sample preparation. Although all these cell lines represent the same subtype of breast cancer, miRNA-overexpressing cells showed distinct results compared to parental MCF7. Surprisingly, there was relatively low separation between the control and H_2_O_2_ treatment groups for all cell lines seen along PC2, consistent with the limited number of significant differentially expressed genes identified in condition-level comparisons (control vs. H_2_O_2_) (Appendix A).

Gene Set Enrichment Analysis (GSEA) was used to assess how each cell line responded to H_2_O_2_. Unsurprisingly, gene sets involved in apoptosis, p53, and ROS signaling were positively enriched, and those associated with cell cycle progression were negatively enriched, relative to the control treatment counterparts (Appendix A). Instead, by comparing each experimental cell line to that of MCF7, we aimed to assess any differences cellular response to H_2_O_2_ exposure with respect to the overexpression of miR-526b, miR-655, or COX-2. We analyzed the top enriched MSigDB hallmark gene sets, as well as the hallmark gene sets of ROS, apoptosis, and the p53 pathway for each MCF7-miR526b (Figure 5A–E), MCF7-miR655 (Figure 5F–J), and MCF7-COX2 (Figure 5K–O) relative to MCF7 following H_2_O_2_ treatment.

In the MCF7-miR526b cell line, the most significant upregulated enrichment was the hallmark spermatogenesis gene set (NES = 1.36, FDR = 0.134) (Figure 5A), and the most significant negative enrichment was the hallmark TNFA signaling via NFKB gene set (NES = −2.21, FDR = 0.0) (Figure 5B). The hallmark apoptosis (NES = −1.8, FDR = 0.001) (Figure 5C) and p53 signaling (NES = −1.61, FDR = 0.004) (Figure 5D) gene sets were negatively enriched, suggesting a weaker apoptotic response and dampened p53 activity in the MCF7-miR526b cell line in response to H_2_O_2_. Similarly, the hallmark ROS gene set was negatively enriched (NES = −1.09, FDR = 1) (Figure 5E), indicating a less pronounced response in ROS-related gene expression relative to MCF7 cells, though this was not significant.

In the MCF7-miR655 cell line, the most significant upregulated gene set was the hallmark interferon alpha response (NES = 1.95, FDR = 0.0) (Figure 5F), suggesting immune response involvement or inflammatory signaling. The hallmark hypoxia gene set was the most significantly downregulated enrichment (NES = −1.59, FDR = 0.075) (Figure 5G), indicating reduced transcription of genes involved in cellular stress response. As with the MCF7-miR526b cell line, the hallmark apoptosis (NES = 1.09, FDR = 0.496) (Figure 5H) and p53 pathway (NES = −1.14, FDR = 0.432) (Figure 5I) gene sets were enriched in the downregulated MCF7-miR655 genes, although not significantly so, indicating that apoptotic signaling remained unaltered relative to MCF7. The hallmark ROS gene set was not significantly enriched in MCF7-miR655 (NES = 0.77, FDR = 1) (Figure 5J).

For MCF7-COX2, the most significant upregulated enrichment was the hallmark E2F targets gene set (NES = 1.89, FDR = 0.001) (Figure 5K), indicating an increase in E2F-driven transcription and cell cycle progression. The most significant downregulated enrichment was the hallmark estrogen response late gene set (NES = −2.09, FDR = 0.0) (Figure 5L), indicating disruption of ER signaling or the activation of alternative signaling pathways. Like MCF7-miR526b, both the apoptosis (NES = −1.32, FDR = 0.069) (Figure 5M) and p53 pathway (NES = −1.1, FDR = 0.34) (Figure 5N) gene sets were negatively enriched in the MCF7-COX2 cell line, indicating reduced transcriptional activation of genes governing stress-induced cell cycle arrest and programmed cell death. Finally, the hallmark ROS gene set was negatively enriched, but this was not statistically significant (NES = −0.85, FDR = 0.821) (Figure 5O).

Analysis of enriched gene ontology (GO) biological processes identified that upregulated DEGs of MCF7-miR526b were enriched in neuronal and synaptic signaling pathways, including synaptic transmission, GABAergic signaling, monoamine and dopamine transport, and chloride ion transport (Figure 5P). These processes suggest activation of neuron-like stress adaptation programs involving membrane excitability and ion channel regulation. Conversely, downregulated DEGs in MCF7-miR526b were associated with suppression of classical stress and growth signaling pathways, including the MAPK/ERK cascade, intracellular signal transduction, inflammatory responses, and cytokine-mediated signaling (Figure 5Q). Developmental processes such as epidermis development and endothelial cell migration were also downregulated, indicating a potential shift away from epithelial stress responses.

In MCF7-miR655 cells, upregulated DEGs were predominantly linked to innate immune responses, including antiviral defense, interleukin-27 signaling, and suppression of viral replication (Figure 5R). Several developmental processes, particularly those associated with eye development and morphogenesis, were also enriched. Meanwhile, downregulated DEGs were strongly associated with extracellular matrix organization, synaptic architecture, and cell–cell adhesion (Figure 5S). Suppressed terms included collagen fibril organization, synapse assembly, and calcium-dependent cell adhesion, indicating a marked loss of epithelial structure and intercellular communication.

In MCF7-COX2 cells, upregulated genes were enriched for signaling and migratory processes, including myoblast migration, tyrosine phosphorylation, postsynaptic membrane organization, and neuron cell–cell adhesion (Figure 5T). Genes involved in metabolic regulation, such as carboxylic acid and lipid metabolism, and negative regulation of PPAR signaling were also elevated. In contrast, downregulated DEGs were associated with epithelial organization and vascular development, including suppression of angiogenesis, epithelial development, axon guidance, and negative regulation of fibroblast growth factor receptor signaling and epithelial cell proliferation (Figure 5U). These patterns suggest a shift toward a migratory, metabolically adaptive, and less differentiated phenotype.

### 2.4. Protein–Protein Interaction (PPI) Network Construction and Hub Gene Analysis

For each set of DEGs, protein–protein interaction (PPI) networks were built using STRING, and these networks were then imported into Cytoscape (v3.10.3) for hub gene analysis (Figure 6). Nodes (genes) were ranked based on connectivity score, which includes degree, betweenness, and closeness centrality, to identify the most influential nodes within each network.

In the MCF7-miR526b upregulated DEGs (Figure 6A), five hub genes were identified: *MAOB*, *SLC6A3*, *SLC18A2*, *H2BC11*, and *CACNA1C* (Figure 6C). These genes are primarily associated with neurotransmitter transport and ion channel activity, aligning with the enrichment of synaptic signaling pathways observed in this condition. For the MCF7-miR526b downregulated set (Figure 6B), the hub genes included *TNF*, *IL18*, *ITGAM*, *STAT4*, and *FCGR2B* (Figure 6C), all of which are key regulators of inflammatory or immune-related processes. Inflammatory genes are downregulated in miR-526b-overexpressed cells.

In the MCF7-miR655 upregulated network (Figure 6D), hub genes included *DHX58*, *MX1*, *OAS2*, *CACNA1C*, and *TRDN* (Figure 6F), many of which are involved in antiviral responses and calcium signaling. In contrast, the MCF7-miR655 downregulated hub genes (Figure 6E), *TNF*, *CALML5*, *GJA1*, *DSG2*, and *A2M* (Figure 6F), highlight suppression of immune and adhesion pathways, consistent with a loss of epithelial characteristics, very similar to the response we recorded with miR-526b.

The MCF7-COX2 upregulated gene network (Figure 6G) identified *CYP2E1*, *GSTM2*, *FGFR1*, *OAS2*, and *IFIT3* as hub genes (Figure 6I), indicating a shift toward metabolic, oxidative, and stress signaling pathways. For the MCF7-COX2 downregulated gene set (Figure 6H), the top-ranking hub genes were *EGFR*, *CD44*, *SOX2*, *CALML5*, and *EGF* (Figure 6I), pointing to repression of stem cell phenotypes, epithelial development, and growth factor signaling.

Together, these hub genes represent central regulators of the distinct transcriptional responses induced by overexpression of miR-526b, miR-655, or COX-2 under oxidative stress and may serve as key nodes for further mechanistic or therapeutic investigation. The downregulation of *CALML5*, identified in both MCF7-miR655 and MCF7-COX2, was validated via qPCR (Appendix A).

### 2.5. Hub Gene Expression in TCGA-BRCA Tissue Samples

Data from The Cancer Genome Atlas (TCGA) was accessed to analyze the expression of each of the identified hub genes. Pathologically normal tissue samples were compared to luminal A, luminal B, HER2-enriched, and basal breast cancer subtypes.

Among the upregulated hub genes identified (Figure 7), most showed the greatest expression in normal tissue and less aggressive breast subtypes than the more aggressive HER2-enriched and basal subtypes. These included *MAOB*, *SLC6A3*, *SLC18*, *DHX58*, *TRDN*, *CYP2E1*, *GSTM2*, *FGFR1*, and *CACNA1C* (Figure 7A–C,E,G–J). Conversely, hub genes such as *H2BC11*, *MX1*, *OAS2*, and *IFIT3* displayed greater expression in more aggressive breast cancer subtypes (Figure 7D,F,M,K).

Of the downregulated hub genes identified (Figure 8), *IL18* was more highly expressed in all breast cancer subtypes relative to normal tissue (Figure 8A), with the highest expression in the basal subtype. Conversely, the expression levels of *STAT4*, *GJA1*, *A2M*, and *EGF* were lowest in aggressive subtypes, such as basal and HER2-enriched, relative to normal tissue and the luminal subtypes (Figure 8C,E,G,I). Other hub genes, such as *EGFR* and *CALML5*, displayed a more complex pattern of gene expression amongst tissue types in which expression in luminal A and luminal B was were lower than normal tissue samples but then rose in the more aggressive HER2-enriched and basal subtypes (Figure 8H,M). Lastly, many displayed only subtle differences in gene expression across tissue types, including *ITGAM*, *FCGR2B*, *DSG2*, *CD44*, *SOX2*, and *TNF* (Figure 8B,D,F,J–L).

### 2.6. Survival Analysis of Hub Genes in Breast Cancer Patients

Overall survival (OS) for breast cancer patients was analyzed with KM plotter for each upregulated (Figure 9) and downregulated (Figure 10) hub gene to identify the prognostic significance of each. Of the hub genes upregulated relative to MCF7, only *OAS2* (Figure 9M) (HR = 1.47, *p* = 0.0026) was significantly associated with poor patient outcome. Other upregulated HUB genes, such as *SLC18A2* (Figure 9C), *MX1* (Figure 9F), and *GSTM2* (Figure 9I), were also linked with poor patient survival, though these findings were not statistically significant. Conversely, many of the upregulated hub genes, such as *MAOB* (Figure 9A) (HR = 0.66, *p* = 0.00026), *SLC6A3* (Figure 9B) (HR = 0.56, *p* = 0.000042), *DHX58* (Figure 9E) (HR = 0.72, *p* = 0.0033), *TRDN* (Figure 9G) (HR = 0.69, *p* = 0.0083), *CYP2E1* (Figure 9H) (HR = 0.76, *p* = 0.026), and *IFIT3* (Figure 9K)) (HR = 0.75, *p* = 0.02), were correlated with improved overall breast cancer patient survival. *FGFR1* (Figure 9J) and *CACNA1C* (Figure 9L) also followed this trend, though not significantly. There was no data available to assess the clinical relevance of *H2BC11* expression and overall patient survival.

Of the downregulated hub genes, only the high expression of *EGFR* (Figure 10H) (HR = 1.35, *p* = 0.02) was associated with poor patient survival. For other hub genes, *IL18* (Figure 10A), *FCGR2B* (Figure 10D), *GJA1* (Figure 10E), *EGF* (Figure 10I), and *SOX2* (Figure 10K), high expression was linked with poor patient survival; however, these results were not significant. Conversely, low expression of many of the identified downregulated hub genes was associated with poor patient survival. These included *ITGAM* (Figure 10B) (HR = 0.77, *p* = 0.029), *STAT4* (Figure 10C) (HR = 0.59, *p* = 0.0000025), *DSG2* (Figure 10F) (HR = 0.62, *p* = 0.000031), *A2M* (Figure 10G) (HR = 0.64, *p* = 0.0025), *CD44* (Figure 10J) (HR = 0.75, *p* = 0.014), *TNF* (Figure 10L) (HR = 0.68, *p* = 0.00082), and *CALML5* (Figure 10M) (HR = 0.72, *p* = 0.0066).

## 3. Discussion

This study aimed to understand how the overexpression of miR-526b, miR-655, and COX-2 influences the cellular response to oxidative stress in breast cancer. Previous research shows that overexpression of miR-526b and miR-655 in breast cancer increases intracellular ROS and SO levels [9]. A feedback loop exists in the tumor microenvironment, where these miRNAs heighten oxidative stress, which, in turn, boost miRNA expression. We hypothesized that this interaction conveys a survival advantage under conditions of oxidative stress. Using H_2_O_2_ to induce oxidative stress, we examined cell viability, DNA damage, and global changes in gene expression. Each experimental cell line demonstrated distinct adaptive response.

MCF7-COX2 cells displayed the most pronounced resilience to H_2_O_2_-induced stress, evidenced by significantly elevated cell viability, reduced PARP cleavage, and lower expression of apoptotic and cell cycle arrest genes. This was supported by negative enrichment of hallmark apoptosis and p53 pathway gene sets and positive enrichment of E2F target genes relative to MCF7, indicative of sustained cell proliferation. These findings align with our hypothesis that COX-2 overexpression, in conjunction with high endogenous expression of miR-526b and miR-655, promotes cell survival under conditions of oxidative stress. The downregulation of PARP cleavage and γ-H2AX intensity further indicates either reduced or sufficiently repaired DNA damage. While total ATM and CHK2 protein levels were largely unchanged, phosphorylated CHK2 did increase, suggesting activation of DNA damage repair, though further investigation into these signaling pathways is warranted as the differences were subtle. Together, this suggests that COX-2 overexpression alters cellular stress responses to favor cell survival and recovery.

MCF7-COX2 also showed an enrichment of metabolic and migration-related processes and suppression of estrogen signaling, angiogenesis, and epithelial maintenance, indicating a metabolically adaptive and survival-oriented response relative to MCF7. Upregulated hub genes such as *CYP2E1*, *FGFR1*, and *GSTM2* suggest metabolic adaptations promoting cell survival and resilience to stress. CYP2E1 is involved in xenobiotic detoxification but is also a driver of ROS production [12]; FGFR1 is linked to proliferation and treatment resistance [13]; and GSTM2 has been identified in antioxidant defense and has been associated with resilience to treatment and OS-induced stress in prostate cancer [14]. Clinically, *OAS2* was associated with poor patient survival in breast cancer, supporting its potential as a stress-adaptive but pro-tumorigenic factor [15,16] and reinforcing the adverse prognostic implications of COX-2. On the other hand, several pro-proliferative and stemness-associated genes (*EGFR*, *EGF*, *CD44*, *SOX2*) were suppressed. Under normal conditions, we have previously shown that CD44 and SOX2 are upregulated in MCF7-COX2 cells [17], making these results of particular interest to us. In some cellular contexts, *CD44* expression can be downregulated during oxidative stress [18], potentially as a protective mechanism to mitigate the effects of excessive reactive oxygen species (ROS), suggesting a more complex and context-dependent relationship, though it has also been shown that CD44 expression and activity can play a critical role in the promotion of antioxidant defenses [19]. EGFR plays a role in cellular antioxidant defenses. Oxidative stress can prevent the normal ubiquitination and subsequent degradation of EGFR, leading to its accumulation in perinuclear compartments, where it may remain active and promote cell survival [20,21]. Alternatively, these results may reflect stress-induced transcriptional reprogramming that compensates or bypasses EGFR signaling through FGFR1 to promote cell survival instead [22,23]. While in theory, reduced EGFR levels may make cells more susceptible to oxidative-stress-mediated damage, future investigation of our model, extending beyond transcriptional activity of EGFR, is required.

MCF7-miR526b cells also showed enhanced viability and reduced apoptotic signaling, albeit to a lesser degree than MCF7-COX2. PARP cleavage was significantly reduced, and p21 protein levels were attenuated following H_2_O_2_ exposure and rest. While p53 protein levels remained elevated, downstream signaling appeared impaired, evident in the significantly negative enrichment of p53 and apoptosis gene sets. These findings suggest a decoupling of p53 expression from its canonical transcriptional activity, consistent with previous reports that show that oxidative-stress-resilient tumors may retain p53 expression but selectively modulate its downstream effectors [24]. This attenuation of apoptotic output may allow cells to better tolerate oxidative insults while avoiding cell death.

Transcriptional analysis of MCF7-miR526b cells revealed suppression of classical stress-related pathways, including MAPK, cytokine signaling, and epithelial development, while upregulated genes were enriched in neuronal and ion signaling, possibly contributing to altered redox regulation via mitochondrial membrane potential and calcium homeostasis. In MCF7-miR526b, *MAOB*, *SLC6A3*, and *CACNA1C* are upregulated, and these markers are linked to metabolism, calcium transport, dopamine signaling, mitochondrial function, ATP production, and redox balance [25,26,27]. MAOB is linked to increased ROS production, which may promote tumor growth in certain instances, and it has paradoxically been associated with better survival outcomes in breast cancer and has also been investigated as a therapeutic target in colon cancer, glioma, and other forms of cancer [28]. Previous studies demonstrated that miR-526b promotes cellular plasticity and metabolic adaptation, with COX-2 playing a key role [29]. Hence, the upregulated markers’ function might contribute to miR-526b-induced oxidative tolerance.

MCF7-miR655 cells, in contrast, displayed a more intermediate phenotype. Neither apoptosis nor p53 signaling were significantly enriched, and PARP cleavage remained comparable to MCF7, indicating that the protective effect of miR-655 is ultimately limited or context-dependent. The modest increase in p21 protein expression post-recovery, despite reduced gene expression, may reflect post-transcriptional regulation or delayed stress responses. Although cell viability and apoptotic markers were not significantly different from MCF7 cells, transcriptomic analysis revealed some distinct patterns of gene expression. Notably, the upregulation of innate immune gene sets, interferon alpha response genes, for example, and the simultaneous downregulation of hypoxia-associated genes suggest that miR-655 may reshape inflammatory and redox-sensitive transcriptional networks. The negative enrichment of hypoxia-related genes conflicts with past findings of MCF7-miR655 cells displaying an enhanced hypoxic response relative to MCF7, where cells were treated with CoCl_2_ [30]. These current results may vary because the observed changes in hypoxic response gene expression might be caused by H_2_O_2_ exposure. Enrichment of antiviral and immune response pathways and downregulation of epithelial and extracellular matrix (ECM) organization genes suggest a shift toward an immune-modulated or mesenchymal phenotype. Hub gene analysis highlighted upregulation of antiviral regulators *OAS2* and *DHX58* and downregulation of epithelial genes such as *DSG2* and *CALML5*, suggesting a shift away from epithelial traits and greater transcriptional activity involving immune pathways.

Additionally, the MCF7 cell line expressed genes linked to neuronal functions, reflecting cancer cell heterogeneity and plasticity—properties similar to embryonic stem cells [31]. We have also previously shown that overexpression of miR-526b, miR-655, and COX-2 promotes the cancer stem cell population and inflammatory responses in breast cancer [10,11,17]. Therefore, the transcriptomic changes observed in neuronal and immunity-related markers in MCF7 cell lines are due to the inherent effects of miR-526b, miR-655, and COX-2 in breast cancer. The enrichment of pathways such as neuronal or developmental signaling may not directly reflect breast cancer biology; thus, we have been cautious not to overstress their relevance, although some studies have indicated that targeting these neural-specific genes in cancer can suppress malignant traits [31].

The paradoxical observation that some hub genes upregulated in our experimental cell lines are associated with a favorable prognosis in breast cancer patients highlights the complexity of cellular stress response and genetic implications that affect disease prognosis. Short-term adaptations that protect cells against acute oxidative stress in vitro may operate within broader signaling networks that suppress tumor progression in vivo. Moreover, the expression of certain genes may be a byproduct of the conditions that necessitate tumorigenesis and thus may display dichotomous roles in cancer.

Our findings suggest therapeutic opportunities, particularly in sensitizing tumors to ROS-inducing treatments by exploiting pathways downstream of miR-526b, miR-655, and COX-2 that either mitigate ROS-induced damage or promote cell survival [2,32]. Ultimately, gene expression cannot substitute for protein expression, and the question of the activity and consequence of the resulting proteins and enzymes must be determined if potential effective treatments are to be actualized. Thus, the function of each of these identified hub genes must be examined.

We also acknowledge the limitations of the model used in our study. The use of MCF7 alone restricts the generalizability of our findings. The MCF7 cell line represents the ER+ tumor subtype, the most diagnosed breast cancer subtype; further work in HER2+ and triple-negative cell lines will be essential. Moreover, DNA damage assays suggest subtle protective effects; they should be interpreted cautiously.

Together, our findings support a mode in which miR-526b, miR-655, and COX-2 enhance resilience to oxidative stress in breast cancer through suppression of apoptosis, modulation of DNA damage response, and transcriptional reprogramming. These adaptations may involve altered metabolism, immune signaling, and epithelial phenotype. The context-dependent expression and survival associations of the identified hub genes highlight potential biomarkers and targets for future investigation into oxidative stress resistance and therapy response in breast cancer.

## 4. Materials and Methods

### 4.1. Cell Culture

Human breast cancer cell line MCF7 was purchased from the American Type Culture Collection (ATCC, Rockville, MD, USA). miR-526b-, miR-655-, and COX-2-overexpressing cell lines were established by plasmid transfection, as previously described [10,11,17]. MCF7-COX2 cells were used as a positive control for both MCF7-miR526b and MCF7-miR655. Cells were cultured in Roswell Park Memorial Institute (RPMI)-1640 media (Gibco, ON, Canada) supplemented with 10% fetal bovine serum (VWR, ON, Canada) and 1% Pen-Strep (Gibco, ON, Canada). Cells were incubated at 37 °C with 5% CO_2_. Transfected cells were maintained with 200 ng/mL Geneticin (Cat. No: G418, Biobasic).

### 4.2. Hydrogen Peroxide Treatment

MCF7 cells were plated into 6-well plates at a density of 100,000 cells per well and given time to attach overnight. Cells were treated with H_2_O_2_ at concentrations of 0, 50, 150, 250, 350, 450, and 550 µM for 48 h. H_2_O_2_ was omitted in the control condition. Cell viability was measured using an automated Corning Cell Counter and repeated for at least three biological replicates. Count values were compared relative to the control group. GraphPad Prism (v10.0.2) (GraphPad Software, Boston, MA, USA) was used to determine the IC50 concentration to be 216 µM. For subsequent experiments, H_2_O_2_ was added to cells in culture, once 70% confluent, at a final concentration of 216 µM for 48 h either with or without a 3-day recovery (rest) period. For the rest period, media with H_2_O_2_ was removed following 48 h treatment, and cells were washed with PBS and given fresh media. Media was then changed again on the second day of the rest period.

### 4.3. Cell Viability

Cells were seeded into 6-well cell culture plates at 100,000 cells per well, and treatment started 12 h later. Following treatment, cells were trypsinized and counted on an automated Countess 3 cell counter. The average of duplicate counts was taken for each of four biological replicates. Viability was assessed as cell number relative to the day of treatment.

### 4.4. RNA Extraction, cDNA Synthesis, and Real-Time qPCR

RNA was extracted using Qiazol and RNeasy Mini Kits (Cat. No: 79306 and 74104, Qiagen, Germantown, MD, USA), and quality was assessed by spectrophotometry. A total of 1 µg of RNA was used for cDNA synthesis with a High-Capacity cDNA Reverse Transcription Kit (Cat. No: 4368813, Thermo Fisher Scientific, Waltham, MA, USA). For quantitative real-time PCR (qRT-PCR), Luna Universal PCR Master Mix (Cat. No: M3004, New England Biolabs, Ipswich, MA, USA) and gene-specific probes (Thermo Fisher Scientific, Waltham, MA, USA) were used. Probes included those for *CDKN1A* (p21, Hs00355782_m1), *PUMA* (BBC3, Hs00248075_m1), *CAT* (Hs00156308_m1), *GPX2* (Hs01591589_m1), *GSR* (Hs00167317_m1), and *CALML5* (Hs00249968_s1). For ΔCT calculations, *RPL5* (Hs03044958_g1) was used as the housekeeping gene. Fold changes were calculated using the 2^−ΔΔCT^ method [33].

### 4.5. SDS-PAGE and Western Blotting

Cells were lysed with cell lysis buffer supplemented with protease and phosphatase inhibitors (Cat. No: 9803 and 5872, New England Biolabs, Ipswich, MA, USA). Protein concentrations were assessed with bicinchoninic acid (BCA) (Cat. No: SK3021, BioBasic, Markham, ON, Canada). A total of 20–30 µg of protein was loaded for SDS-PAGE electrophoresis using 10% (p53, Chk2/p-Chk2, PARP, AKT/p-AKT), 12% (p21), and 4–15% gradient (ATM) SDS–polyacrylamide gels. Transfers were carried out onto a nitrocellulose membrane (0.2 µm pore size) (Cat. No: 10600007, Cytiva, Uppsala, Sweden). Primary antibodies for proteins of interest included p21 (Cat. No: 2947, Cell Signaling Technology, Beverly, MA, USA), p53 (Cat. No: sc-126, Santa Cruz Biotechnology, Dallas, TX, USA), ATM (Cat. No: 2873, Cell Signaling Technology, Beverly, MA, USA), Chk2 (Cat. No: 13954-1-AP, Proteintech, Rosemont, IL, USA), p-Chk2 (Thr68) (Cat. No: 29012-1-AP, Proteintech, Rosemont, IL, USA), PARP (Cat. No: 9542S, Cell Signaling Technology, Beverly, MA, USA), AKT (Cat. No: 9272S, Cell Signaling Technology, Beverly, MA, USA), and p-AKT (Ser473) (Cat. No: 9271S, Cell Signaling Technology, Beverly, MA, USA). Primary antibodies were incubated in dilutions of 1:500 for p21, p53, AKT, and p-AKT and 1:200 for ATM, Chk2, p-Chk2, and PARP at 4 °C overnight with agitation. Blots were incubated with anti-ACTB (Cat. No: sc-47778, Santa Cruz Biotechnology, Dallas, TX, USA), anti-Tuba1 (Cat. No: Y054861, Applied Biological Materials, Vancouver, BC, Canada), or anti-vinculin (Cat. No: sc-73614, Santa Cruz Biotechnology, Dallas, TX, USA) primary antibodies at 1:1000 dilutions, serving as housekeeping proteins. Blots were washed and then incubated with Azurespectra 700 and Azurespectra 800 (Cat. No: AC2129 and AC2134, Azure Biosystems, Dublin, CA, USA) NIR-fluorescent secondary antibodies at a 1:10,000 dilution for 1 h at room temperature. The dry membranes were imaged using an Azure Sapphire Biomolecular Imager (Azure Biosystems) and analyzed with the AzureSpot software (v2.2.167), applying rolling-ball background noise removal and housekeeping gene normalization. Graphs illustrate fold changes in normalized fluorescent band volume (normalized volume) relative to untreated samples. AKT phosphorylation was analyzed as the ratio of phosphorylated AKT to total AKT. The cleavage of PARP was analyzed as the ratio of cleaved PARP (~89 kDa fragment) to total PARP.

### 4.6. Single-Cell Gel Electrophoresis (Comet Assay)

Comet assays were performed with an Abcam Comet Assay Kit (Cat. No: ab238544, Abcam, Cambridge, UK) according to the manufacturer’s protocol. Electrophoresis was performed in an alkaline running buffer at 26 V (1 V/cm) and 300 mA for 25 min. Fluorescent images were taken on a Nikon Eclipse Ti-2 microscope at 4× and 10× magnification. Comet metrics (tail DNA%, tail moment, and olive moment) were analyzed via the ImageJ (v1.53k) plugin OpenComet (v1.3.1) [34]. Twenty images were analyzed per sample. Means for each statistic were calculated over three biological replicates.

### 4.7. Immunocytochemistry (ICC)

Immunocytochemistry was performed for phosphorylated H2A histone family member X (γ-H2AX (S139)). Cells were washed twice with ice-cold PBS and fixed with methanol for 10 min at −20 °C. Following fixation, cells were washed and blocked with 1% BSA in PBS for 1 h at room temperature. Fixed samples were incubated in primary γ-H2AX (S139) antibody (Cat. No: 9718, Cell Signaling Technology, Beverly, MA, USA) at a 1:800 dilution in 0.1% BSA in PBS overnight at 4 °C with slow agitation. Samples were washed and incubated with TRITC-conjugated secondary antibody (Cat. No: T-2769, Thermo Fisher Scientific, Waltham, MA, USA) at a dilution of 1:1000 in 0.1% BSA in PBS for 1 h at room temperature. Cells were washed three times with PBST (0.05% Tween) and mounted onto coverslips with DAPI-supplemented mounting medium (Cat. No: Ab104139, Abcam, Cambridge, UK). Slides were sealed, stored at 4 °C, and imaged the following day. Fluorescent imaging was carried out on the Nikon Eclipse Ti-2 at 40× magnification. Staining was analyzed with the NIS Elements AR software (v5.42.01) by limiting the region of interest for quantification to the nucleus of each cell within a given image, providing an average fluorescent intensity for each image. At least 10 images were analyzed per sample.

### 4.8. RNA-Sequencing and High-Throughput Data Pipeline

RNA-sequencing was performed at the Centre d’expertise et de services Génome Québec (CES) (McGill University, Montréal, QC, Canada). As samples were extracted as part of a larger study, based on the response to other chemotherapeutic drugs, samples were treated with either H_2_O_2_ or DMSO as the control for 48 h. Following treatment, cells were resuspended in Qiazol (Qiagen, MD, USA) and frozen before shipment on dry ice. mRNA library preparation with polyA enrichment was performed before sequencing with a NovaSeq 6000 sequencing system (PE100—25M reads). Sequence quality was assessed with FastQC (v 0.11.9) [35] and trimmed with Trimmomatic (v0.39) [36]. Sequences were aligned and counted with the Spliced Transcripts Alignment to a Reference (STAR) (v2.7.11b) [37] and High-throughput Sequence (HTSeq) packages (v2.0.8) [38] against the GRCh38.p14 reference genome. Count files were analyzed for differential gene expression with the Bioconductor package, DESeq2 (v1.46.0) [39], in RStudio (v2024.12.0.467, R v4.4.2). Log2 fold changes were computed, and multiple testing correction was performed using the Benjamini–Hochberg method to control for the false discovery rate (FDR). This data is represented graphically in volcano plots generated in RStudio (v2023.12.0 + 369) using the ggplot2 (v3.5.1) [40] package. Genes with an adjusted *p*-value less than 0.05 and log2 fold changes above 1.5 or below −1.5 were considered differentially expressed genes (DEGs) and included in downstream analysis. This data can be found in the Gene Expression Omnibus (GEO) database [41] under the series record GSE295502.

### 4.9. Gene Set Enrichment Analysis (GSEA)

Gene set enrichments were assessed with the Gene Set Enrichment Analysis (GSEA) software (v4.3.3) [42]. Hallmark (H) Molecular Signature Database (MSigDB) gene sets were analyzed. Differentially expressed genes were ranked by the signal-to-noise metric, and normalized enrichment scores (NES) were calculated after 1000 gene set permutations. Gene sets with an FDR under 0.25 were considered significantly enriched.

### 4.10. Analysis of Enriched Biological Processes

DEGs were classified based on upregulation and downregulation. These lists were submitted to the Enrichr web server [43] to evaluate enriched biological processes (BPs). The raw data for these enrichments were downloaded from Enrichr and re-plotted to show the top 10 biological processes, colored by adjusted *p*-value, along with related gene counts and gene ratios (the ratio of gene count to the total number of genes defined by a given biological process).

### 4.11. Network and Hub Gene Analysis

Upregulated and downregulated DEGs were submitted to STRING [44] to generate protein interaction networks. The full network type, where edges represented both functional and physical protein interactions, was selected with a high confidence interaction score of 0.7. The network was exported to Cytoscape (v3.10.3) [45], and the cytoHubba plugin (v0.1) [46] was used to evaluate network topology. The degree, betweenness, and closeness methods were applied to identify the most connected nodes (genes) within each network. The top 5 genes, based on combined topological scores from these three methods, were considered hub genes for each particular network.

### 4.12. Kaplan–Meier (KM) Survival Analysis

Overall survival analysis based on gene expression was conducted using the online survival analysis tool *KM plotter* [47]. The RNA-seq mRNA dataset was used, and the analysis included all breast cancer patients. Patients were categorized based on the optimal expression cutoff value determined by the KM plotter algorithm.

### 4.13. Accession of the Cancer Genome Atlas (TCGA)—Breast Cancer (BRCA) Patient Data

Transcriptomic breast cancer data was obtained from the Cancer Genome Atlas (TCGA) [48] via the TCGAbiolinks R package (v2.31.1) [49]. Patients were stratified by normal tissue and the four basic breast cancer subtypes based on their PAM50 classification [50]. Expression values in transcripts per million (TPM) were transformed (log_2_(TPM + 1)) and assessed for each gene of interest.

### 4.14. Statistical Analysis

Statistical analysis and graph creation were carried out with GraphPad Prism Software (v10.4.0). Cell viability (Figure 1C) was evaluated using one-way ANOVA followed by Dunnett’s post hoc test for multiple comparisons. This approach also assessed differences in gene expression among cell lines (Figure 1D,E). One-way ANOVA followed by Tukey’s post hoc test was used to assess treatment differences in protein expression (Figure 1G,H and Figure 3G–I). Two-way ANOVA followed by Tukey’s post hoc test analyzed cell line and treatment impacts on AKT phosphorylation (Figure 1I), PARP cleavage (Figure 1J), tail DNA percentage, tail moment, and olive moment (Figure 2B–D), as well as differences in immunofluorescent cell staining (Figure 3E). Differential gene expression was analyzed with the default parameters of the DESeq2 package (v1.38.3) (Figure 4A–C). Count data were normalized using the median-of-ratios method, which determines a size factor for each sample based on the median of ratios of raw counts to the geometric mean across all samples. Gene-wise dispersion estimates were modelled under a negative binomial distribution with empirical Bayes shrinkage. Log2 fold changes were derived from the fitted model to measure differences between conditions or cell types, and the Wald test determined statistical significance. The Benjamini–Hochberg method was used to calculate adjusted *p*-values, controlling the false discovery rate. *p*-values below 0.05 were considered significant. For TCGA gene expression data, normality was presumed due to large sample sizes. Differences between sample types were assessed with the non-parametric Kruskal–Wallis test followed by Dunn’s multiple comparisons test (Figure 7 and Figure 8). A *p*-value of less than 0.05 was considered statistically significant.

## 5. Conclusions

Overexpression of COX-2 provided the greatest protection against oxidative damage, significantly reducing apoptosis and enhancing the expression of genes critical for cell cycle progression. Cells overexpressing miR-526b and miR-655 exhibited distinct yet overlapping stress responses, such as decreased DNA damage responses and altered p53 signaling pathways. High-throughput transcriptome analysis revealed key hub genes involved in redox homeostasis, immune response, and metabolism—many of which are linked to breast cancer patient survival. These findings highlight the potential of miR-526b, miR-655, and COX-2 as pivotal factors in enhancing resistance to oxidative stress through transcriptional reprogramming, emphasizing their potential roles as promising biomarkers or therapeutic targets that warrant further validation, as additional research is needed to elucidate the precise mechanisms involved.

## Figures and Tables

**Figure 1 ijms-26-09103-f001:**
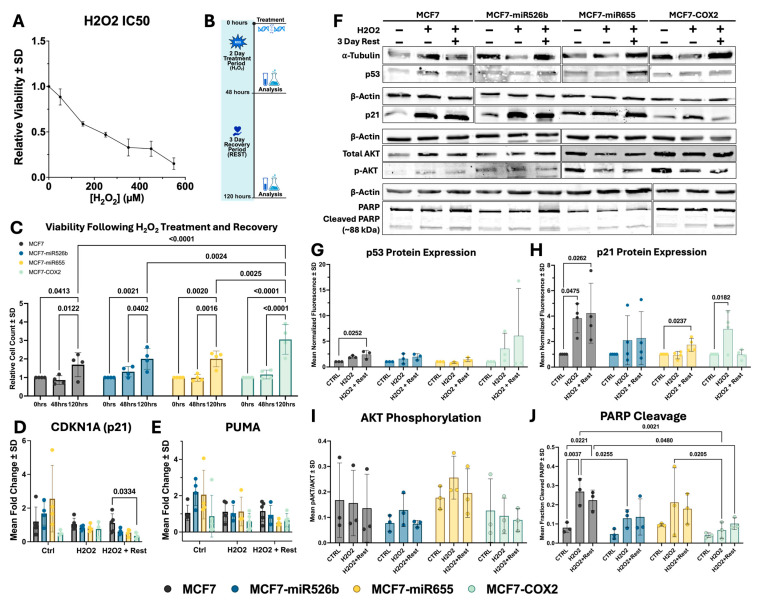
Analysis of cell viability after H_2_O_2_ exposure. (**A**) Dose–response curve for H_2_O_2_-treated MCF7. (**B**) Timeline of treatment and recovery. Created with BioRender.com. (**C**) Viability of MCF7, MCF7-miR526b, MCF7-miR655, and MCF7-COX2 cells after treatment. Data represented as mean +/− SD, (one-way ANOVA, Dunnett’s post hoc). (**D**) *CDKN1A* (*p21*) and (**E**) *PUMA* gene expression. Data shown as mean +/− SD (one-way ANOVA, Dunnett’s post hoc). (**F**) Western blot images and quantification for (**G**) p53, (**H**) p21, (**I**) AKT phosphorylation (Ser473), and (**J**) PARP cleavage (~89 kDa). Plus (+) and minus (−) symbols represent the presence or absence of H_2_O_2_ and 3 days of rest. Full blot images added in the Appendix A. Data presented as mean +/− SD (one-way ANOVA with Tukey’s post hoc).

**Figure 2 ijms-26-09103-f002:**
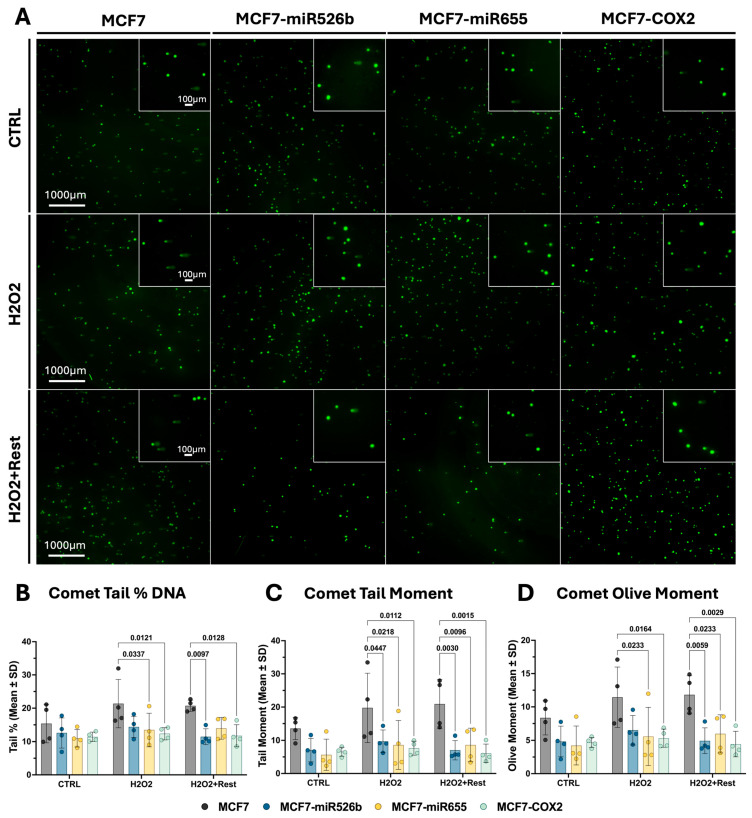
H_2_O_2_-induced DNA damage assessed via the comet assay. (**A**) Comet assay images from each cell line in control, H_2_O_2_, and rest conditions. Images taken at 4× magnification, and the scale bar is 1000 µm. Insets were imaged at 20× magnification, and the scale bar is 100 µm. Quantification of (**B**) tail DNA percentage, (**C**) tail moment, and (**D**) olive moment. Data shown as mean +/− SD (two-way ANOVA with Tukey’s post hoc).

**Figure 3 ijms-26-09103-f003:**
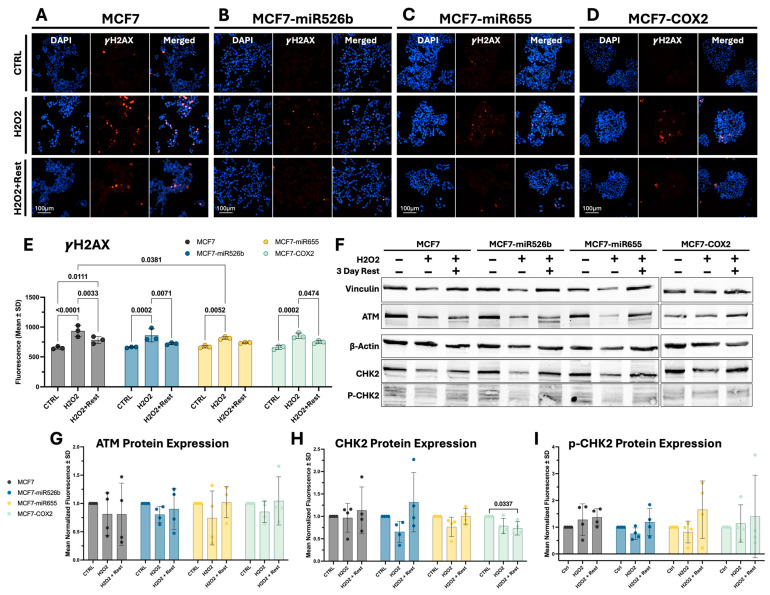
γ-H2AX (S139) staining and DNA damage response protein expression. Fluorescent images of γ-H2AX staining for (**A**) MCF7, (**B**) MCF7-miR526b, (**C**) MCF7-miR655, and (**D**) MCF7-COX2 cells following H_2_O_2_ treatment and rest. Images taken at 40× magnification, and the scale bar is 100 µm. Red, γ-H2AX; blue, DAPI. (**E**) Quantification of γ-H2AX staining. Data shown as mean +/− SD (two-way ANOVA with Tukey’s post hoc). (**F**) Western blot images of ATM, Chk2, and p-Chk2 following H_2_O_2_ treatment and rest. Plus (+) and minus (−) symbols represent the presence or absence of H_2_O_2_ and 3 days of rest. Full blot images can be found in the Appendix A. Quantification of Western blot analysis for (**G**) ATM, (**H**) Chk2, and (**I**) p-Chk2. Data shown as mean +/− SD (one-way ANOVA with Tukey’s post hoc).

**Figure 4 ijms-26-09103-f004:**
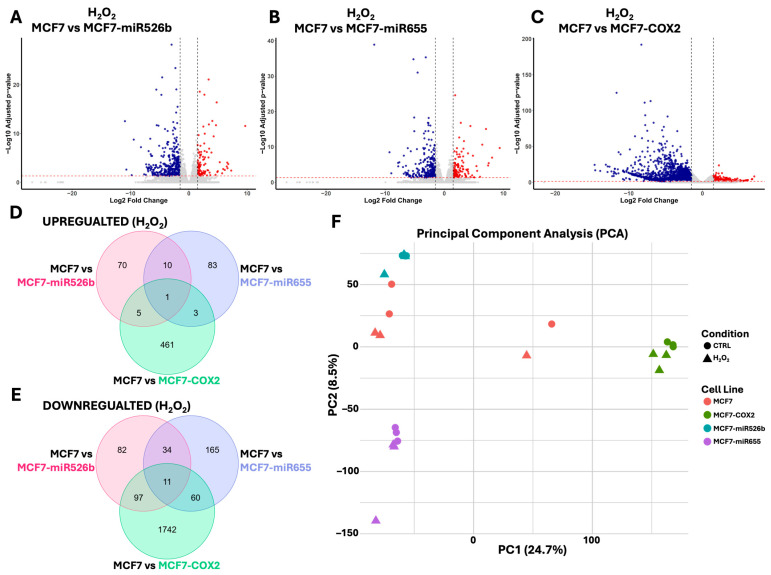
RNA-sequencing and differentially expressed genes following H_2_O_2_ treatment. Volcano plots assessing differentially expressed genes in either (**A**) MCF7-miR526b, (**B**) MCF7-miR655, or (**C**) MCF7-COX2 cell lines relative to MCF7 after H_2_O_2_ treatment. Log2 fold change cutoffs are under −1.5 (blue) or above 1.5 (red) and adjusted *p*-values below 0.05. Significantly (**D**) upregulated and (**E**) downregulated DEGs were compared, and (**F**) principal component analysis (PCA) was used to analyze sample-to-sample variability (circle = CTRL and triangle = H_2_O_2_ treatment; red = MCF7, teal = MCF7-miR526b, blue = MCF7-miR655, and green = MCF7-COX2).

**Figure 5 ijms-26-09103-f005:**
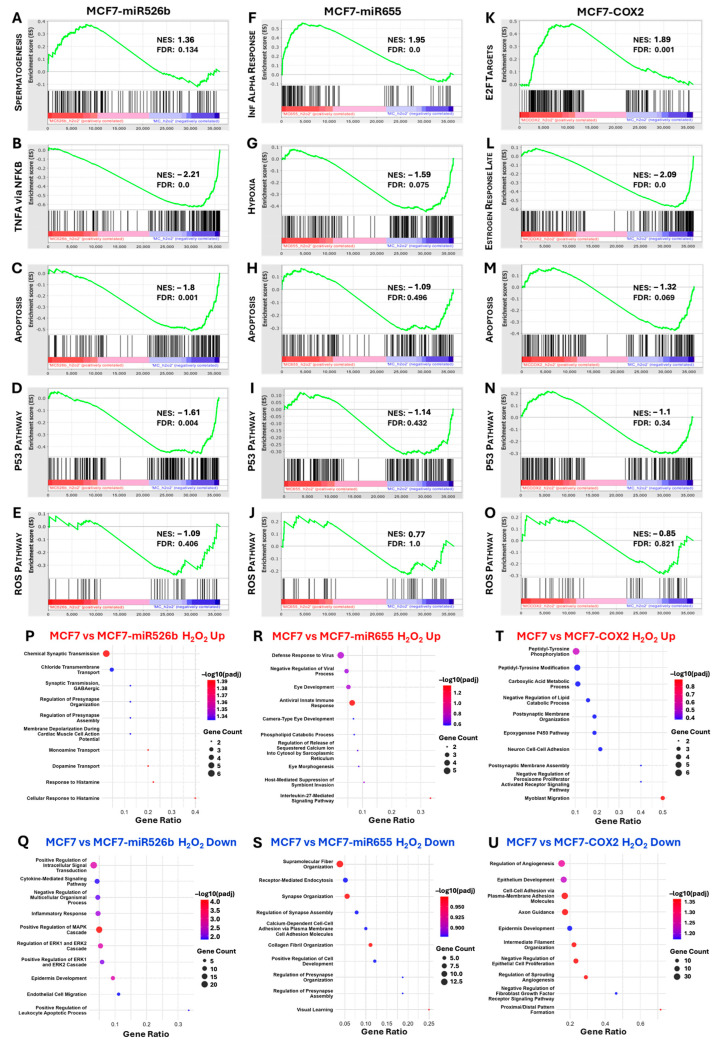
RNA-sequencing and enriched hallmark gene sets and biological processes. GSEA enrichment plots show the most significant upregulated and downregulated enrichments as well as the enrichments of the hallmark apoptosis, p53 pathway, and ROS pathway gene sets in either MCF7-miR526b, MCF7-miR655, or MCF7-COX2 cell lines relative to MCF7 after H_2_O_2_ treatment. For MCF7-miR526b, (**A**) the most significant positively enriched gene set was that of spermatogenesis, while (**B**) the most significant negatively enriched gene set was TNFA signaling via NFKB. The (**C**) apoptosis, (**D**) p53 pathway, and (**E**) ROS pathway gene sets were all negatively enriched. For MCF7-miR655, (**F**) the most significant positively enriched gene set was that of interferon alpha response, while (**G**) the most significant negatively enriched gene set was hypoxia. The (**H**) apoptosis and (**I**) p53 pathway gene sets were negatively enriched, while the (**J**) ROS pathway gene set was positively enriched. For MCF7-COX2, (**K**) the most significant positively enriched gene set was that of E2F targets, while (**L**) the most significant negatively enriched gene set was estrogen response late. The (**M**) apoptosis, (**N**) p53 pathway, and (**O**) ROS pathway gene sets were all negatively enriched. NES is the normalized enrichment score, and FDR is the false discovery rate. Dot plots show the top ten most significantly enriched biological processes for the upregulated (red text) and downregulated (blue text) DEGs in either (**P**,**Q**) MCF7-miR526b, (**R**,**S**) MCF7-miR655, or (**T**,**U**) MCF7-COX2 cell lines relative to MCF7 after H_2_O_2_ treatment. Dot plots are sorted by gene ratio, colored by −log(padj), and scaled by gene count.

**Figure 6 ijms-26-09103-f006:**
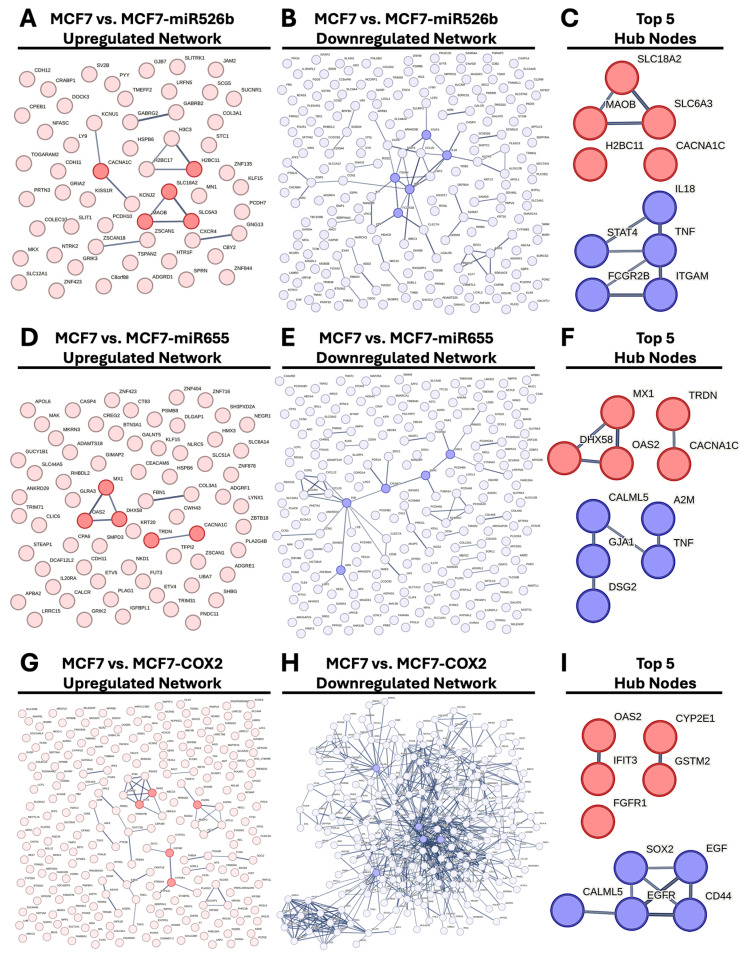
Cell line comparison of H_2_O_2_ response and identification of network hub genes. Venn diagrams comparing (**A**) upregulated and (**B**) downregulated DEGs between each cell line after H_2_O_2_ treatment. Dot plots depicting the top ten enriched biological processes associated with the (**C**) 473 common upregulated DEGs and (**D**) 324 common downregulated DEGs. Interaction networks of (**E**) upregulated and (**F**) downregulated DEGs. (**G**) The top 5 hub genes from each network. Heat maps display the expression of each identified hub gene in each experimental cell line relative to MCF7 in (**H**) control and (**I**) H_2_O_2_ treatment conditions.

**Figure 7 ijms-26-09103-f007:**
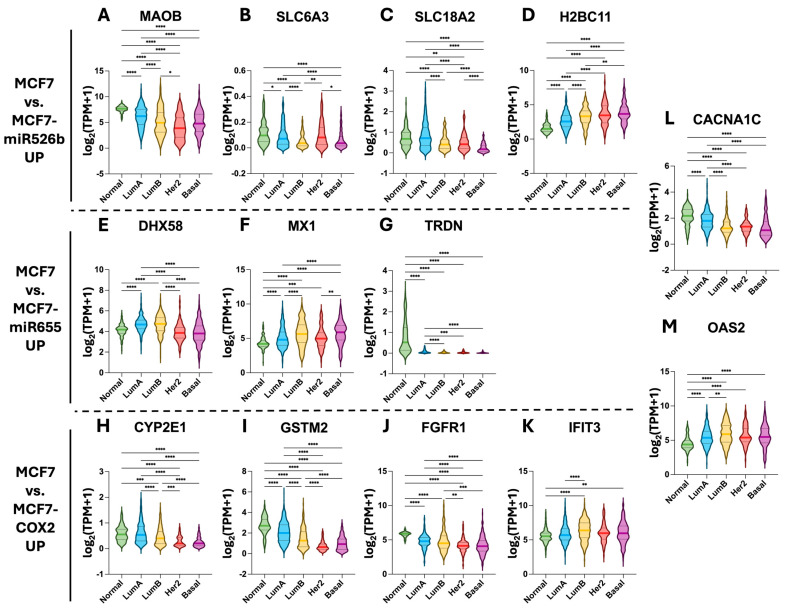
Analysis of upregulated hub gene expression in breast cancer subtypes. (**A**–**D**) Log2(TPM + 1) expression of the upregulated hub genes in MCF7-miR526b relative to MCF7 after H_2_O_2_ exposure. (**E**–**G**) Log2(TPM + 1) expression of the upregulated hub genes in MCF7-miR655 relative to MCF7 after H_2_O_2_ exposure. (**H**–**K**) Log2(TPM + 1) expression of the upregulated hub genes in MCF7-COX2 relative to MCF7 after H_2_O_2_ exposure. Shared upregulated hub genes between (**L**) MCF7-miR526b and MCF7-miR655, as well as (**M**) MCF7-miR655 and MCF7-COX2. Tissue types include normal patient tissue (*n* = 113), as well as luminal A (lumA, *n* = 571), luminal B (lumB, *n* = 209), HER2-enriched (Her2, *n* = 82), and basal (*n* = 197) subtypes. Data sourced from TCGA, shown as violin plots, medians denoted with solid lines, and first quartiles and third quartiles denoted with dotted lines (* *p* < 0.05, ** *p* < 0.01, *** *p* < 0.001, **** *p* < 0.0001, Kruskal–Wallis’s test, followed by Dunn’s multiple comparisons).

**Figure 8 ijms-26-09103-f008:**
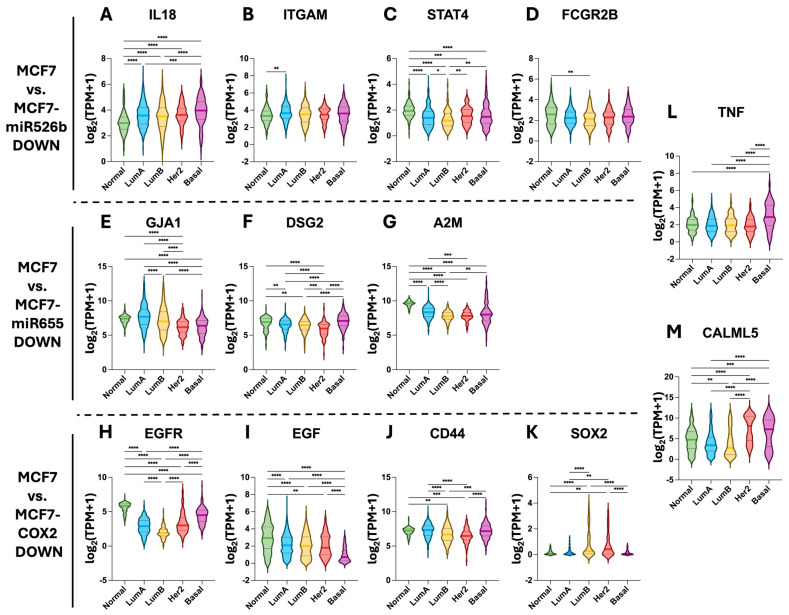
Analysis of downregulated hub gene expression in breast cancer subtypes. (**A**–**D**) Log_2_(TPM + 1) expression of the downregulated hub genes in MCF7-miR526b relative to MCF7 after H_2_O_2_ exposure. (**E**–**G**) Log_2_(TPM + 1) expression of the downregulated hub genes in MCF7-miR655 relative to MCF7 after H_2_O_2_ exposure. (**H**–**K**) Log_2_(TPM + 1) expression of the downregulated hub genes in MCF7-COX2 relative to MCF7 after H_2_O_2_ exposure. Shared downregulated hub genes between (**L**) MCF7-miR526b and MCF7-miR655, as well as (**M**) MCF7-miR655 and MCF7-COX2. Tissue types include normal patient tissue (*n* = 113), as well as luminal A (lumA, *n* = 571), luminal B (lumB, *n* = 209), HER2-enriched (Her2, *n* = 82), and basal (*n* = 197) subtypes. Data sourced from TCGA, shown as violin plots, medians denoted with solid lines, and first quartiles and third quartiles denoted with dotted lines (* *p* < 0.05, ** *p* < 0.01, *** *p* < 0.001, **** *p* < 0.0001, Kruskal–Wallis’s test, followed by Dunn’s multiple comparisons).

**Figure 9 ijms-26-09103-f009:**
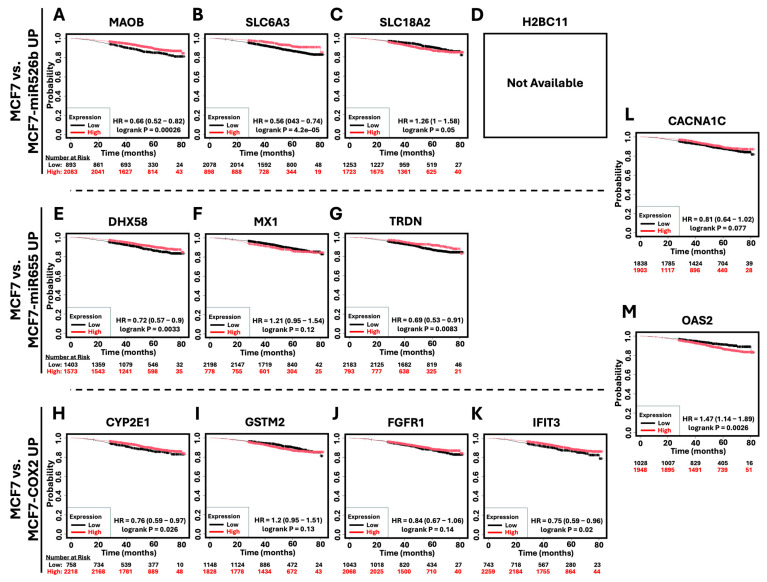
Kaplan–Meier survival analysis of hub genes in breast cancer patients. Overall survival analysis based on the expression of the identified upregulated hub genes in breast cancer patients. (**A**–**D**) Overall survival based on the expression of the upregulated hub genes in MCF7-miR526b relative to MCF7 after H_2_O_2_ exposure (*MAOB*, *SLC6A3*, and *SLC18A2*). Data for H2BC11 was not available. (**E**–**G**) Overall survival based on the expression of the upregulated hub genes in MCF7-miR655 relative to MCF7 after H_2_O_2_ exposure (*DHX58*, *MX1*, and *TRDN*). (**H**–**K**) Overall survival based on the expression of the upregulated hub genes in MCF7-COX2 relative to MCF7 after H_2_O_2_ exposure (*CYP2E1*, *GSTM2*, *FGFR1*, and *IFIT3*). Overall survival based on the expression of shared upregulated hub genes between (**L**) MCF7-miR526b and MCF7-miR655 (*CACNA1C*), as well as (**M**) MCF7-miR655 and MCF7-COX2 (*OAS2*).

**Figure 10 ijms-26-09103-f010:**
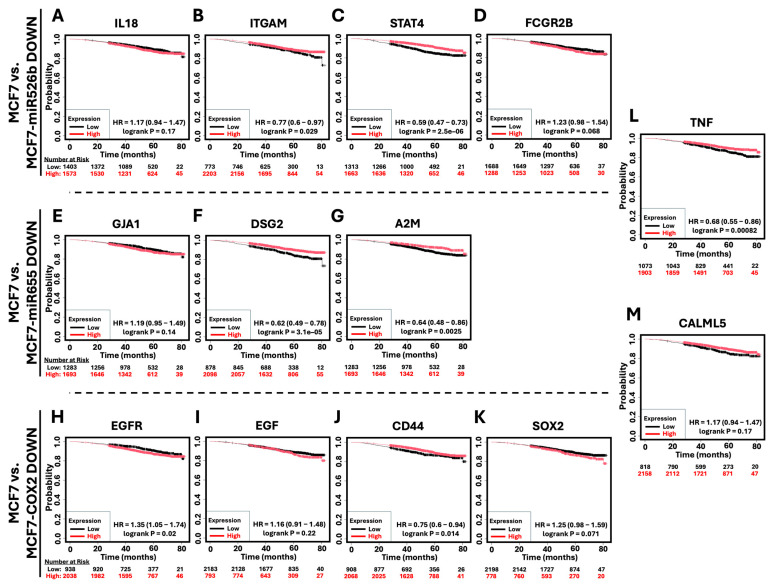
Kaplan–Meier survival analysis of hub genes in breast cancer patients. Overall survival analysis based on the expression of the identified downregulated hub genes in breast cancer patients. (**A**–**D**) Overall survival based on the expression of the downregulated hub genes in MCF7-miR526b relative to MCF7 after H_2_O_2_ exposure (*IL18*, *ITGAM*, *STAT4*, and *FCGR2B*). Data for *H2BC11* was not available. (**E**–**G**) Overall survival based on the expression of the downregulated hub genes in MCF7-miR655 relative to MCF7 after H_2_O_2_ exposure (*GJA1*, *DSG2*, and *A2M*). (**H**–**K**) Overall survival based on the expression of the downregulated hub genes in MCF7-COX2 relative to MCF7 after H_2_O_2_ exposure (*EGFR*, *EGF*, *CD44*, and *SOX2*). Overall survival based on the expression of the shared downregulated hub genes between (**L**) MCF7-miR526b and MCF7-miR655 (*TNF*), as well as (**M**) MCF7-miR655 and MCF7-COX2 (*CALML5*).

## Data Availability

The supporting information is available for download here—Appendix A. The raw data supporting the conclusions of this article will be made available by the authors on request. Sequencing data has been deposited in the GEO database (https://www.ncbi.nlm.nih.gov/geo/, accessed on 24 April 2025) under the series record GSE295502.

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
