# Peer review of "Distinct Oxidative Stress Adaptations Driven by the Overexpression of miR-526b, miR-655, and COX-2 in Breast Cancer"

_ijms, 2025, doi:10.3390/ijms26189103_

Round 1

Reviewer 1 Report

Comments and Suggestions for Authors

In the following work, Opperman et al. They evaluated how overexpression of miR-526b, miR-655, and COX-2 influences the response of breast cancer cells to H₂O₂-induced oxidative stress.

Minor points

  1. Regarding the authors' conclusions on the regulation of apoptosis (Figure 1), I believe that they should evaluate some other factor involved in the apoptosis process in order to be able to conclude on this matter (caspases, for example).
  2. The references to the axes of the different graphs in Figure 5 are not clearly read.

Author Response

In the following work, Opperman et al. They evaluated how overexpression of miR-526b, miR-655, and COX-2 influences the response of breast cancer cells to H₂O₂-induced oxidative stress.

Response: Thank you very much for giving us the opportunity to revise our manuscript. All changes are highlighted in the manuscript.

Minor points

  1. Comment: Regarding the authors' conclusions on the regulation of apoptosis (Figure 1), I believe that they should evaluate some other factor involved in the apoptosis process in order to be able to conclude on this matter (caspases, for example).

Response: Thank you for your insights. While we agree that assessing additional markers such as caspases would provide more information, our study already includes analysis of several apoptotic regulators and markers. We assessed p53 protein expression, as well as p21 gene and protein expression, PUMA (p53 upregulated modulator of apoptosis) gene expression, an early marker of apoptosis and PARP (Poly (ADP‐ribose) polymerase), an active marker of apoptosis at a later stage. Cleavage of PARP, which indicates active caspases, is a prominent characteristic of apoptosis. Please see figures 1E and 1F, page 3. Furthermore, our high-throughput RNA-sequencing revealed a significant positive enrichment of apoptotic pathway genes (See Supplementary Figure S9), providing transcriptomic evidence as well.

We have conducted an additional experiment to validate hub gene expression. Please see supplementary figure S10.

  1. Comment: The references to the axes of the different graphs in Figure 5 are not clearly read.

Response: We thank the reviewer for this helpful feedback. Figure 5 has been updated. We enlarged the text for the references to the y-axis to make it easier to read. Please see page number 8.

Reviewer 2 Report

Comments and Suggestions for Authors

Reviewer Comment (for submission)_ JMS_ijms-3844225

Opperman et al., “Distinct Oxidative-Stress Adaptations Driven by the Overexpression of miR-526b, miR-655, and COX-2 in Breast Cancer”

The manuscript entitled “Distinct Oxidative-Stress Adaptations Driven by the Overexpression of miR-526b, miR-655, and COX-2 in Breast Cancer” investigates the role of oncogenic microRNAs and COX-2 in mediating breast cancer cell adaptation to oxidative stress. Using H₂O₂ exposure models in MCF7 parental and engineered cell lines, the authors assess viability, DNA damage, transcriptomic changes, and hub gene relevance in TCGA datasets. The integration of RNA-seq, GSEA, PPI hub-gene analysis, and survival correlations provides a comprehensive dataset. The study addresses an important topic—how tumor cells adapt to oxidative stress, a mechanism underlying therapeutic resistance—and offers potentially novel biomarkers. However, while the study is ambitious and the datasets are extensive, several methodological, interpretational, and conceptual issues limit the current impact. In particular, some RNA-seq enrichments are difficult to reconcile with breast cancer biology, validation is limited, and clinical translation remains underdeveloped. Addressing these points will substantially strengthen the manuscript.

Major Comments

  1. Conceptual novelty vs. incremental advance

The study largely confirms prior knowledge, and the authors should clarify the novel contribution by presenting a clearer mechanistic model, specifying whether the effects of miR-526b/miR-655 and COX-2 under oxidative stress are primarily mediated through DNA repair, apoptosis suppression, metabolic adaptation, or immune modulation.

  1. RNA-seq interpretation requires biological plausibility

Several enriched pathways (e.g., neuronal or antiviral signaling) appear unrelated to breast cancer biology and may reflect noise or artifacts. The authors should provide a clear biological rationale or supporting evidence to avoid overstating their relevance.

  1. Validation of transcriptomic findings

RNA-seq findings lack experimental validation. Key DEGs or hub genes (e.g., MAOB, OAS2, EGFR) should be confirmed by qRT-PCR, Western blot, or independent datasets to ensure reproducibility.

  1. Survival analysis inconsistencies

Some hub genes linked to better prognosis are paradoxically upregulated in stress-adapted cells, creating a disconnect with clinical data. The authors should address this contradiction and integrate it more critically into the discussion.

  1. Choice of model system (MCF7)

Using only the luminal MCF7 cell line limits generalizability. Including or at least acknowledging the lack of more aggressive subtypes (e.g., TNBC, HER2+) would strengthen the study.

  1. DNA-damage and repair assays

DNA damage assays show only subtle differences, so claims of protection should be tempered. ATM/Chk2 results are inconclusive and should be discussed more cautiously.

Minor Comments

  1. Clarity of figures and supplementary data

Some western blots (e.g., Figure 1F for PARP, p53) are difficult to interpret, and higher-quality images or densitometry would improve clarity.

  1. Statistical presentation

Provide exact p-values or effect sizes and avoid over-interpreting non-significant findings.

  1. Gene ontology enrichment

Filter out developmentally irrelevant GO terms (e.g., eye morphogenesis) to focus on biologically relevant results.

  1. Language and style

Condense overly descriptive sections and correct minor grammatical errors.

  1. Clinical translation

Better connect findings to therapeutic applications, such as sensitizing tumors to ROS-inducing therapies.

Conclusion

This study addresses an important question of oxidative stress adaptation in breast cancer, but its impact is limited by insufficient novelty, lack of validation, and weak clinical translation. To strengthen the manuscript, the authors should (i) clarify the novel mechanistic contribution, (ii) provide biological rationale for RNA-seq enrichments, (iii) experimentally validate key hub genes, (iv) address contradictions in survival analysis, (v) acknowledge model limitations, and (vi) temper claims on DNA damage assays. Improving figure clarity, statistical rigor, and clinical relevance will further enhance the manuscript.

Author Response

Manuscript ID: ijms-3844225

TITLE: Distinct oxidative-stress adaptations driven by the overexpression of miR-526b, miR-655, and COX-2 in breast cancer

AUTHORS: Reid Morgan Opperman, Sujit Maiti, Mousumi Majumder

Reviewer 2

Reviewer Comment (for submission)_ JMS_ijms-3844225

Opperman et al., “Distinct Oxidative-Stress Adaptations Driven by the Overexpression of miR- 526b, miR-655, and COX-2 in Breast Cancer”

Comment: The manuscript entitled “Distinct Oxidative-Stress Adaptations Driven by the Overexpression of miR-526b, miR-655, and COX-2 in Breast Cancer” investigates the role of oncogenic microRNAs and COX-2 in mediating breast cancer cell adaptation to oxidative stress. Using H₂O₂ exposure models in MCF7 parental and engineered cell lines, the authors assess viability, DNA damage, transcriptomic changes, and hub gene relevance in TCGA datasets. The integration of RNA-seq, GSEA, PPI hub-gene analysis, and survival correlations provides a comprehensive dataset. The study  addresses  an  important  topic—how  tumor  cells  adapt  to  oxidative  stress,  a  mechanism underlying therapeutic resistance—and offers potentially novel biomarkers. However, while the study is ambitious and the datasets are extensive, several methodological, interpretational, and conceptual issues limit the current impact. In particular, some RNA-seq enrichments are difficult to  reconcile  with  breast  cancer  biology,  validation  is  limited,  and  clinical  translation  remains underdeveloped. Addressing these points will substantially strengthen the manuscript.

Response: Thank you for your insights and for giving us an opportunity to revise the manuscript. We have now conducted additional experiments to address your concern. Please note all changes highlighted in yellow in the revised manuscript.

Major Comments

  1. Conceptual novelty vs. incremental advance

Comment: The study largely confirms prior knowledge, and the authors should clarify the novel contribution by presenting a clearer mechanistic model, specifying whether the effects of miR-526b/miR-655 and COX-2 under oxidative stress are primarily mediated through DNA repair, apoptosis suppression, metabolic adaptation, or immune modulation.

Response: Thank you. You are correct that there are many questions that remain unanswered in the current study. We present here new differential outcomes to oxidative stress mediated through miR-526b, miR-655 and COX2 overexpression, including damaged DNA, alterations in apoptotic responses, changes in metabolism, or immune modulation. High-throughput analysis of miR-526b, miR-655 and COX-2 overexpressed breast cancer cells exposed to H2O2 has never been done before, which is the novelty we present. A few of the common candidate genes differentially expressed in miRNA-high cells become potential therapeutic targets in future studies (such as OAS2, TNF, CACNA1C, CALML5). This study lays the foundation for further investigating the roles of miRNAs and COX2 under oxidative stress to determine a clear mechanistic model.

  1. RNA-seq interpretation requires biological plausibility

Comment: Several enriched pathways (e.g., neuronal or antiviral signaling) appear unrelated to breast cancer biology and may reflect noise or artifacts. The authors should provide a clear biological rationale or supporting evidence to avoid overstating their relevance.

Response: We thank the reviewer for the critique. As you suggested, it is possible that some of the identified enriched pathways may not be entirely relevant to breast cancer. We have updated the discussion to prevent overstating the relevance of these results (See page 18, lines 544-548). Additionally, the MCF-7 breast cancer cell line expresses some genes that are also found in neurons, particularly genes related to synaptic transmission and neuronal generation, as we observed in our study. This unexpected finding may be due to the inherent heterogeneity and plasticity of cancer cells, which can exhibit properties of embryonic neural precursor cells (J Biol Chem, 2017 Jun 20;292(31):12842–12859. doi: 10.1074/jbc.M117.785865). We have now discussed that on page 18 lines 529-538.  

  1. Validation of transcriptomic findings

Comment: RNA-seq findings lack experimental validation. Key DEGs or hub genes (e.g., MAOB, OAS2, EGFR) should be confirmed by qRT-PCR, Western blot, or independent datasets to ensure reproducibility.

Response: We thank the reviewer for this comment. For the time being, we were able to obtain a qPCR probe for one of the hub genes (CALML5), though we agree that validation of all hub genes must be a focus of future studies. We chose to prioritize CALML5 as it is not well known and poorly understood in breast cancer and was identified in two of the three experimental cell lines. Please see the revised Supplementary Figure S10 where we confirm the downregulation of this gene following H2O2 exposure relative to MCF7, as observed from the sequencing results. See page 19, line 599 for the updated materials and methods.

  1. Survival analysis inconsistencies

Comment: Some hub genes linked to better prognosis are paradoxically upregulated in stress- adapted cells, creating a disconnect with clinical data. The authors should address this contradiction and integrate it more critically into the discussion.

Response: We agree with the reviewer's comments to some degree and appreciate their insight, which highlights an intriguing issue regarding how certain gene expressions affect stress response and patient outcomes. We acknowledge that some hub genes upregulated in transfected cell lines are paradoxically associated with better patient outcomes, which might seem contradictory—indeed, it may be. Nonetheless, we think this also illustrates the complexity and context-dependent nature of gene expression and function in biology.

1) Firstly, our results suggest that overexpressing miR-526b, miR-655, and COX-2 provides some natural protection against oxidative stress, specifically from hydrogen peroxide. However, this protection appears to be a short-term response, not a sign of long-term resistance developed through prolonged stress exposure. While these genes may be temporarily upregulated and help cells survive in vitro under oxidative conditions, this doesn't necessarily mean they contribute to more aggressive tumor behavior in patients. Some genes that help cells cope with stress might also be involved in larger signaling networks that actually limit disease progression. So, even if a gene boosts short-term survival in lab settings, it doesn't always correlate with worse outcomes in patients.

2) Secondly, the Kaplan-Meier results include all breast cancer subtypes and stages from diverse individuals with different genetic backgrounds, health conditions, and environmental factors. This also explains why we agree with your next point (comment #5) regarding the limitations of the model system.

3) Finally, there is the ever-present possibility of a disconnect between the expression of a gene and the activity of its protein counterpart. A marker’s differential expression might be linked with breast cancer; however, we identified these markers' differential expression in the context of H2O2 exposure. Hence, the markers’ function needs to be explained in miRNA overexpressed cells in oxidative stress-induced conditions. This molecular determination goes beyond the scope of the current study but will be essential for any future work.

To conclude, while we agree that it would seem logical for genes upregulated in a more aggressive cell line to be correlated with poor patient outcome, the data does not support this in every case – which was part of the rational for performing such a test. An objective of this study was to identify possible genes of interest and observing their effects in patient populations was used as a filter to guide us in future studies and direct our efforts towards genes with biological relevance and clinical implications. We have addressed this comment in a broader context in the updated discussion (See page 18, lines 539-545).

  1. Choice of model system (MCF7)

 Comment: Using only the luminal MCF7 cell line limits generalizability. Including or at least acknowledging the lack of more aggressive subtypes (e.g., TNBC, HER2+) would strengthen the study.

Response: We thank the reviewer for their comment. We previously showed a possible interaction between miR526b and miR655 and p53 (Tordjman J, 2019), and miRNA promotes oxidative stress in MCF7 cells (Shin B, 2019). Given the wild-type status of p53 in MCF7, we chose this cell line to explore this possible interaction between oxidative stress and miRNA. Although our current findings showed an indirect interaction with p53 and miRNA in the context of oxidative stress. Regardless, MCF7 is an ER+ breast cancer cell line, which accounts for the majority of breast cancer cases. Given this fact, we thought it relevant to study MCF7, though we fully agree that further exploration of different cell lines and breast cancer subtypes should be investigated in future studies. We have acknowledged this limitation in the manuscript (See page 18, lines 553-556)

  1. DNA-damage and repair assays

Comment: DNA damage assays show only subtle differences, so claims of protection should be tempered. ATM/Chk2 results are inconclusive and should be discussed more cautiously.

Response: We thank the reviewer for the comment and have updated our discussion for these results accordingly. (See page 4, line 146; page 17, line 459; and page 18, lines 556-557)

Minor Comments

  1. Clarity of figures and supplementary data

 Comment: Some western blots (e.g., Figure 1F for PARP, p53) are difficult to interpret, and higher- quality images or densitometry would improve clarity.

Response: We appreciate the reviewer's suggestion. We have improved the quality in figure 1F.

  1. Statistical presentation

 Comment: Provide exact p-values or effect sizes and avoid over-interpreting non-significant findings.

Response: Exact p-values have been added to figures 1, 2 and 3.

  1. Gene ontology enrichment

 Comment: Filter out developmentally irrelevant GO terms (e.g., eye morphogenesis) to focus on biologically relevant results.

Response: We appreciate the reviewer's insightful feedback. We've organized the GO terms by how common they are (gene count) and their statistical significance. While we understand that developmental terms like “eye morphogenesis” might seem unrelated, in cancer research, these processes could also indicate regulation of cell movement or an EMT phenotype, though this remains theoretical. Our goal in examining enriched biological processes was to gain a broader understanding of our DEG networks and the cellular functions potentially involved in stress responses. We also want to acknowledge that selectively including certain terms might unintentionally introduce bias, emphasizing their importance in cancer, which we wish to avoid—though we value your perspective. With your approval, we would request that the GO terms remain, though we plan to limit their discussion.

  1. Language and style

Comment: Condense overly descriptive sections and correct minor grammatical errors.

Response: We thank the reviewer for the feedback. We agree that some sections can be dense and disrupt the flow of the article. We have done our best to address this concern throughout. We have also made an effort to correct grammatical errors.

  1. Clinical translation

Comment: Better connect findings to therapeutic applications, such as sensitizing tumors to ROS- inducing therapies.

Response: We agree, thank you for the feedback. We have attempted to address this in the discussion (See page 18, lines 544-550)

Conclusion

Comment: This study addresses an important question of oxidative stress adaptation in breast cancer, but its impact is limited by insufficient novelty, lack of validation, and weak clinical translation. To strengthen the manuscript, the authors should (i) clarify the novel mechanistic contribution, (ii) provide biological rationale for RNA-seq enrichments, (iii) experimentally validate key hub genes,

(iv) address contradictions in survival analysis, (v) acknowledge model limitations, and (vi) temper claims on DNA damage assays. Imp al rigor, and clinical relevance will further enhance the manuscript.

Response: Thank you, we have now addressed each point you listed here.

Round 2

Reviewer 2 Report

Comments and Suggestions for Authors

The authors have made substantial improvements in response to the previous review comments. The introduction, methods, and results are now clearer, and the figures are overall well presented. The revised manuscript is much improved and addresses most of the earlier concerns. Only a few minor points remain for consideration. The English could be polished slightly for clarity and conciseness. Some figure panels, particularly the RNA-seq enrichment plots, remain dense, and adding a short summarizing statement in the figure legends would improve readability. In addition, the interpretation of neuronal and immune pathway findings should be presented cautiously, as the authors have already acknowledged. Overall, this is a solid revision, and I believe the manuscript is suitable for publication after minor editorial adjustments.

Author Response

Comments and Suggestions for Authors

The authors have made substantial improvements in response to the previous review comments. The introduction, methods, and results are now clearer, and the figures are overall well presented. The revised manuscript is much improved and addresses most of the earlier concerns. Only a few minor points remain for consideration. The English could be polished slightly for clarity and conciseness. Some figure panels, particularly the RNA-seq enrichment plots, remain dense, and adding a short summarizing statement in the figure legends would improve readability. In addition, the interpretation of neuronal and immune pathway findings should be presented cautiously, as the authors have already acknowledged. Overall, this is a solid revision, and I believe the manuscript is suitable for publication after minor editorial adjustments.

Responses: Thank you very much. We have tried our best to consider each of your comments, change the figure legend, and improve the quality of all figures, including RNA-seq plots. Please see all changes in the manuscript highlighted.

These are all the changes made to this new draft:

  • We reworded the legend of Figure 5 (see pages 9 and 10, lines 230-242; and page 10, lines 245-246).
  • We added additional clarity to the methods (4.10. Analysis of Enriched Biological Processes, see pages 22 and 23, lines 698-700).
  • We enlarged the size and improved the quality of all figures (please see pages 3, 5, 6, 7, 9, 12, 14, 15, 16 and 17, lines 78, 154, 167, 194, 229, 325, 369, 392, 419 and 441).